# Glucocorticoid-Regulated Kinase CAMKIγ in the Central Amygdala Controls Anxiety-like Behavior in Mice

**DOI:** 10.3390/ijms232012328

**Published:** 2022-10-14

**Authors:** Marcin Piechota, Urszula Skupio, Małgorzata Borczyk, Barbara Ziółkowska, Sławomir Gołda, Łukasz Szumiec, Klaudia Szklarczyk-Smolana, Wiktor Bilecki, Jan Manuel Rodriguez Parkitna, Michał Korostyński

**Affiliations:** 1Department of Molecular Neuropharmacology, Maj Institute of Pharmacology, Polish Academy of Sciences, 12 Smętna Str., 31-343 Kraków, Poland; 2Department of Neurobiology and Neuropsychology, Institute of Applied Psychology, Jagiellonian University, 4 Łojasiewicza Str., 30-348 Kraków, Poland

**Keywords:** Camk1g, glucocorticoids, glucocorticoid receptor, anxiety, amygdala, fear-conditioning, gene expression

## Abstract

The expression of the Calcium/Calmodulin-Dependent Protein Kinase I gamma (encoded by the Camk1g gene) depends on the activation of glucocorticoid receptors (GR) and is strongly regulated by stress. Since Camk1g is primarily expressed in neuronal cells of the limbic system in the brain, we hypothesized that it could be involved in signaling mechanisms that underlie the adaptive or maladaptive responses to stress. Here, we find that restraint-induced stress and the GR agonist dexamethasone robustly increase the expression of Camk1g in neurons of the amygdalar nuclei in the mouse brain. To assess the functional role of Camk1g expression, we performed a virally induced knock-down of the transcript. Mice with bilateral amygdala-specific Camk1g knock-down showed increased anxiety-like behaviors in the light-dark box, and an increase in freezing behavior after fear-conditioning, but normal spatial working memory during exploration of a Y-maze. Thus, we confirm that Camk1g is a neuron-specific GR-regulated transcript, and show that it is specifically involved in behaviors related to anxiety, as well as responses conditioned by aversive stimuli.

## 1. Introduction

Environmental stressors elicit a whole range of physiological and behavioral responses. Although these responses aim to adapt the organism to stress, at times the reaction to stress results in pathology. Disorders with onset associated with stressful events are debilitating and include post-traumatic stress disorder, anxiety disorders, obsessive compulsive disorder, major depression, and other mood disorders [1,2,3]. A therapeutic target that could help to prevent the translation of stress into anxiety and other psychiatric conditions would be of great benefit to the society.

The biological mechanisms involved in the translation of stress responses into neuropathology are only partially known. After a stressful situation, the hypothalamic–pituitary–adrenal (HPA) axis becomes activated, and glucocorticoids (GCs) are released. GCs activate the glucocorticoid receptor (GR), which acts as a transcription factor. The HPA axis dysregulation is implicated in the etiopathology of stress-related psychiatric disorders [4,5]. GR is expressed in the limbic system of the brain including the amygdala, which is considered to be the central brain region involved in the translation of stress into anxiety [6,7,8]. Importantly, GR signaling in the amygdala plays a protective role in order to mitigate the effects of stress [9]. GR presence in the central amygdala (CeA) is also necessary for the conditioning of fear responses [10]. In this region, GR is expressed both in neurons and in astrocytes, and depressed human subjects have been shown to have higher GR levels in the amygdala [11]. As a transcription factor, GR controls the expression of a number of genes in the brain [12]. Still, individual molecular factors that could be responsible for translating its effects into behavioral states are yet to be determined.

Here, we investigate a gene that could serve as a link between GR and anxiety-related behaviors. Camk1g is predominantly expressed in neurons of the limbic system including within the amygdala [13]. Our group has previously shown that its transcription in neurons is triggered by glucocorticoids and that this gene is not expressed in glial cells [14]. We postulate that Camk1g may be involved in the neuroadaptive action of glucocorticoids released in response to stress. This hypothesis is further supported by the fact that CaMKIγ was shown to induce CREB phosphorylation and gene expression in cells, including brain neurons, and therefore can act as a cell transcription program regulator downstream of the GR [15,16]. In this study, we investigated the specific transcriptional regulation of Camk1g in the amygdala after both acute stress and treatment with the GR agonist dexamethasone. Next, the effect of Camk1g knockdown in the mouse amygdala on anxiety-related phenotypes was studied.

## 2. Results

### 2.1. Camk1g Distribution, Regulation and Induction after Stress

Our group has previously shown that transcription of Camk1g in neurons is triggered by glucocorticoids and that this gene is not expressed in glial cells [14]. Here, we examined the spatial pattern of GR-dependent Camk1g expression in the mouse forebrain. This analysis was performed using in situ hybridization in the brains of control and dexamethasone (DEX)-injected animals (Figure 1a). In agreement with a previous report [16], high expression of Camk1g was observed in control animals only in the central nucleus of the amygdala (CeA) and the ventromedial hypothalamic nucleus (VMH). A much lower expression level was detected in the CeA efferent structure, the bed nucleus of stria terminalis (BNST) [17], as well as in the hippocampus and throughout the cortex (most distinctly in the medial frontal cortex).

Following DEX administration, the Camk1g gene was markedly up-regulated in the CeA, while less pronounced changes were observed in the VMH and BNST (Figure 1b). Interestingly, we also found a striking Camk1g gene induction in the lateral septum and the striatum, where it was particularly strong in the olfactory tubercle, the medial part of the caudate/putamen, and the nucleus accumbens core (Figure 1a).

The CeA has been long-known to be involved in mediating behavioral effects of restraint stress [18]. In order to uncover the possible role of the Camk1g in the adaptation of mice to stress, we proceeded to investigate the role of this gene in the amygdala. In the in situ hybridization, we showed that GR stimulation induced Camk1g expression in the CeA (Figure 1a,b). However, it was not known if stress would have similar effects. Here, we reviewed microarray profiling performed by our group [19] and sought to detect Camk1g mRNA levels in the amygdala following a session of restraint stress. We observed an upregulation of Camk1g expression in the amygdala region after a single session of restraint stress (Figure 1c).

### 2.2. Behavioral Consequences of Camk1g Knockdown in CeA

Next, we aimed to investigate the behavioral role of Camk1g in the CeA. We focused on CeA as it is one of the brain centers known to regulate physiological and behavioral responses to stress including adaptation [20,21,22], it was one of the regions where DEX remarkably stimulated Camk1g expression, and restraint stress upregulated Camk1g mRNA in the amygdala. To knock-down (KD) Camk1g locally, we used an shRNA-mediated gene silencing approach delivered using lentiviral particles. First, we established the transfection efficiency using primary neuronal cultures at around 90% with a viral titer of 2.7 × 10^7^/μL (Appendix A). Then, we evaluated the efficiency of Camk1g KD with qPCR and determined that the shRNA targeted towards Camk1g lowered the expression of this gene by 59% as compared with a control shRNA (Appendix A). For in vivo experiments, shRNA-containing lentiviral particles were injected bilaterally into the CeA of adult male mice. An example representation of transgene expression is presented in Appendix A. We further confirmed the effectiveness of Camk1g KD at the protein level with an immunofluorescent approach (Appendix A), where Camk1g-KD animals had markedly downregulated overlap between the Camk1g protein signal and GFP signal as compared with control animals (6% for Camk1g-KD vs. 28% for controls). We, therefore, concluded that the shRNA efficiently lowers Camk1g expression in the CeA.

The amygdala plays a central role in anxiety-like responses to emotionally arousing situations, and stress hormones acting through the GR are thought to be one of the pivotal effectors in this process [23]. Here, we have assessed the behavior of CeA Camk1g knockdown (CeA-Camk1g-KD) mice in a series of tests that measure stress-induced memory formation and expression as well as anxiety-like behavior and working memory.

To measure stress-induced memory formation and expression, we performed a fear conditioning test using foot shocks (5 × 0.4 mA) paired with a tone as aversive stimuli. Both CeA-Camk1g-KD and control mice showed a gradual increase in freezing after each consecutive foot shock delivered during fear conditioning training (time effect: F4,96 = 37.66, *p* < 0.0001), indicating that learning occurred in both groups. However, CeA-Camk1g-KD mice presented overall higher freezing reactions when compared with their control littermates (genotype effect: F1,24 = 6.387, *p* = 0.0185) (Figure 2a). Post hoc analysis using the Bonferroni test indicated that the only point with a significant difference was after the fourth consecutive shock (t120 = 2.652, *p* < 0.05).

We then measured conditioned fear expression in two subsequent retrieval sessions. When compared to control mice, significant increase in freezing was detected in CeA-Camk1g-KD mice (F1,24time 6.472, *p* = 0.0178; F1,24genotype = 5.409, *p* = 0.0288; F1,24int = 0.9707, *p* = 0.3343) (Figure 2b). Post hoc analysis using the Bonferroni test indicated that the difference was significant during the first retrieval session (t48 = 2.458, *p* < 0.05). What is more, CeA-Camk1g-KD mice spent less time in the light area of the light-dark box, which is aversive to anxious mice (t24 = 2.522, *p* = 0.0187). Latency to enter the light compartment remained unchanged (t24 = 1.405, *p* = 0.1729) (Figure 2c,d).

Finally, to additionally control for the potential effect of CeA-Camk1g-KD on memory, we performed a Y maze test. In the Y maze, the frequency of entering the least recently visited arm is an indication of a functioning working memory. In the test, we did not detect any differences in working memory between CeA-Camk1g-KD mice and control mice (t23 = 1.046, *p* = 0.3066) (Figure 2e).

## 3. Discussion

Here, we have used a multiple-technique approach, including shRNA-mediated gene silencing, to show the role of Camk1g expression in stress-related phenotypes. This resounds with our previous findings which showed neuronal changes in gene expression after pharmacological disruptions of the central nervous system function [14,24,25]. CaM kinases were previously identified as crucial factors for neuroplasticity. For example, the calcium-calmodulin-dependent kinase II (CaMKII) is critical for the early phase of LTP. Thanks to the autophosphorylation mechanism, CaMKII can sustain its activity far longer than the initial pulse of calcium, and further activate downstream processes resulting in synaptic change [26]. The next prominent example is calcium-calmodulin-dependent kinase IV (CaMKIV) which is an important activator of CREB transcription factor [27]. Inhibition of CaMKIV impairs the late phase of LTP [28]. CaMKIγ has been recently identified as one of the central genes of a co-expressed gene module regulated in schizophrenia; thus, it may also be involved in other affective disorders [29]. Although CaMKIγ is often overlooked among other calcium-dependent kinases, our study for the first time shows a key role of this protein kinase in the action of glucocorticoids in stress.

First, we decided to inspect the spatial distribution of Camk1g in the central nervous system. In agreement with the initial report by Takemoto-Kimura et al. [16], using in situ gene expression analysis of Camk1g, we found that its basal expression is restricted to the limbic system, including subregions of the extended amygdala, hippocampus, hypothalamus, and the medial frontal cortex. The highest Camk1g expression was observed in a group of interconnected structures: CeA, BNST and the VMH. CeA is a major output nucleus of the amygdala, whose projections to the brainstem and hypothalamus (part of them via the BNST [17]) mediate conditioned fear responses such as freezing, autonomic arousal and stress hormones release [20,30,31]. In turn, the VMH, receiving inputs from the olfactory system via the medial amygdala and BNST [32,33,34], plays a major role in the innate defensive reactions to threat such as a predator, as well as in sexual behaviors [35,36]. The abundant expression of Camk1g in such a set of brain regions alone suggests the potential impact of this kinase on emotion processing, particularly on the stress-related conditioned and unconditioned fear.

Therefore, we inspected the influence of glucocorticoids and stress on Camk1g gene expression. The effects of glucocorticoids on the central nervous system are well known [37]; however, molecular pathways through which these effects are carried out are still not widely understood. A synthetic glucocorticoid, dexamethasone (DEX), activated Camk1g in multiple brain areas. We found a conspicuous induction of the Camk1g gene by DEX in the medial parts of the striatum and in the lateral septum, which resembled that observed in our previous study after morphine administration [25].

Among the structures with a high basal expression of the gene, the CeA showed the most pronounced change in the Camk1g transcript level. Glucocorticoid hormones-induced molecular alterations within the CeA, such as Camk1g up-regulation, may be considered as a candidate mechanism of stress-elicited emotional disturbances, taking into account the major role that CeA plays in the expression of fear and stress reactions [21,30,38]. For that reason, we focused our further efforts on this particular nucleus. The DEX activation of Camk1g together with a previously published microarray dataset from the amygdala of mice subjected to restraint stress [19], show that the amygdalar Camk1g is up-regulated not only by exogenous GR agonist administration but also as a result of physiological stress.

Finally, we aimed to investigate the role of CAMKIγ in the CeA in the stress response. Despite major limitations, the fear conditioning procedure used in the study is still regarded as a model for the stress-related pathogenesis of anxiety disorders [39]. Animals with Camk1g knockdown displayed higher freezing during acquisition and in the first response to a cue. Furthermore, we used the light-dark box test, which is a model allowing to screen for unconditioned anxiety responses in rodents. Camk1g knockdown animals presented higher anxiety in this test.

In the central nervous system, CAMKs play crucial roles in several neural processes, such as mobilization of synaptic vesicles, modulation of ion channels, regulation of gene expression, regulation of muscle contraction, and LTP [40]. Although not much is known about substrate specificity of particular CAMKs, there are studies that suggest at least some degree of their selectivity. For example, activity-dependent cap-dependent translation, but not cap-independent translation, is sensitive to knockdown of CaMKIγ and CaMKIβ but not to knockdown of CaMKIα, CaMKIδ, or CaMKII [41]. Through this mechanism, glucocorticoids could enable activity-dependent translation in particular neuronal circuits.

Our data present a coherent picture of CaMKIγ involvement in neural adaptations after stress. The stress involves an increase in the level of glucocorticoids. This recruitment of the hypothalamic–pituitary–adrenal axis results in the modulation of downstream circuits, such as neuronal pathways in the limbic system. This modulation is manifested by the regulation of Camk1g. Similarly to the suppressive effects of glucocorticoids on the immune system or suppressive effects on AVP release during hemorrhage, here the glucocorticoids prevent the “overshooting” of a stress response. How this overshoot affects further glucocorticoid levels in the absence of or during stress requires further investigation. Nevertheless, without Camk1g, freezing after stress is higher. This ultimately shows that glucocorticoids not only increase stress effects but can also prevent dysfunctional and maladaptive responses.

## 4. Materials and Methods

### 4.1. Animals

Adult male (10 to 14 weeks old at the beginning of experiments) C57BL/6J mice (Jackson Laboratory, Bar Harbor, ME, USA) were housed 7 to 10 per cage under a 12 h dark/light cycle (lights on at 7 a.m.) with free access to food and water. Animals weighing 21 to 31 g were used throughout the experiments. All animal experiments were performed in accordance with the Directive 2010/63/EU of the European Parliament and of the Council of 22 September 2010 on the protection of animals used for scientific purposes, and were approved by the II Local Bioethics Commission (Kraków, Poland; permission number 84/2018).

### 4.2. Dexamethasone Administration and In Situ Hybridization

The animals were sacrificed 4 h after the i.p. injection with 16 mg/kg dexamethasone phosphate (DEX; Dexaven, JELFA S.A., Jelenia Góra, Poland) or saline. After sacrifice, the brains were removed and frozen on dry ice. They were then cut into 12 μm thick coronal sections on a cryostat microtome CM 3050S (Leica Microsystems, Nussloch, Germany), and the sections were thaw-mounted on gelatin-chrome alum-coated slides and processed for in situ hybridization. The hybridization procedure was performed as previously described [42]. Briefly, the sections were fixed with 4% paraformaldehyde, washed in PBS and acetylated by incubation with 0.25% acetic anhydrite (in 0.1 M triethanolamine and 0.9% sodium chloride). The sections were dehydrated using increasing concentrations of ethanol (70 to 100%), treated with chloroform for 5 min and rehydrated with decreasing concentrations of ethanol. The sections were hybridized for 15 h at 37 °C with an oligonucleotide probe complementary to Camk1g cDNA (cggtgcagggctgtgtttccatcaatccaggggtgtctgagggc; accession No.: NM_144817.2). The probes were labeled with 35S-dATP (Hartmann Analytic, Braunschweig, Germany) by the 3’-tailing reaction using terminal transferase (MBI Fermentas, Vilnius, Lithuania). After hybridization, the slices were washed three times for 20 min with 1 × SSC (0.15 M sodium chloride, 0.015 M sodium citrate)/50% formamide at 40 °C and twice for 50 min with 1 × SSC at room temperature. Then, the slices were dried and exposed to phosphorimager plates (Fujifilm, Tokyo, Japan). The hybridization signal was digitized using a Fujifilm BAS-5000 phosphorimager and Image Reader software (Fujifilm). Where appropriate, the ISH signal was quantified using the MCID Elite system (Imaging Research, St. Catharines, Ontario, Canada). Mean signal density, expressed in photostimulated luminescence units/mm2, was measured in selected brain regions. The background signal was measured over the corpus callosum and was subtracted from the hybridization signal in the regions of interest.

### 4.3. Restraint Stress and Microarray Analysis

The whole-genome profiling of animals after restraint stress has been published previously [19] and the data are available in GEO [GSE74002]. Briefly, a procedure [43] was used to induce an uncontrollable aversive situation that produces both physical and psychological acute stress. The animals were put for 30 min into a plastic cylindrical tube with breathing holes and an aperture in the cap for the tail. After the restraint procedure, the animals were returned to their cages. Each animal was exposed to a single restraint session. The animals were sacrificed 2 h after the restraint session. Samples containing all amygdalar nuclei were dissected. RNA was isolated according to the manufacturer’s protocol and was further purified using the RNeasy Mini Kit (Qiagen Inc., Valencia, CA, USA). A starting amount of 200 ng high-quality total RNA was used to generate cDNA and cRNA using the Illumina TotalPrep RNA Amplification Kit (Illumina Inc., San Diego, CA, USA). Each cRNA sample (1.5 μg) was hybridized overnight to MouseWG-6 v2 BeadChip arrays (Illumina) in a multiple-step procedure according to the manufacturer’s instructions; the chips were washed, dried, and scanned on the BeadArray Reader (Illumina). Raw microarray data were generated using BeadStudio v3.0 (Illumina). A total of 10 Illumina MouseWG-6 v2 microarrays were used for this analysis. Microarray analysis and quality control was performed using the BeadArray R package.

### 4.4. Cloning and Lentiviral Vectors Generation

shRNA targeting Camk1g Plasmid (pLKO5) encoding shRNA cassette targeting mouse Camk1g was purchased from Sigma Aldrich (St. Louis, MO, USA; clone ID: TRCN0000361994). During the amplification step of the shRNA Camk1g DNA fragment, SalI/XbaI cloning sites were added. Subsequently, the amplified DNA was cloned into LeGO-G plasmid (Addgene, Watertown, MA, USA; Cat. No.: #27347). To produce the lentiviral vectors, the LeGO-G plasmids encoding shRNA sequences: control (shRNA targeting nonmammalian turbo GFP protein) or targeting the Camk1g were used. Titers of both viruses were comparable and ranged between 1.13 × 108 and 2.19 × 108 transducing U/mL. Virus suspensions were stored at −70 °C until use and were kept on ice immediately before injection.

### 4.5. Primary Cultures

The ex vivo experiments were conducted using mouse striatal neurons and astrocytes from primary cultures [44]. Neuronal tissues were dissected from mouse embryos on 17th/18th day of gestation [45]. The tissues were separately cut into small pieces and digested with trypsin (0.1% for 15 min at RT). The cells were suspended in Neurobasal^®^ medium (Gibco Thermo Fisher, Waltham, MA, USA) supplemented with B27 supplement (Gibco Thermo Fisher) and plated onto polyornithine-coated multiwell plates (TPP, Trasadingen, Switzerland). This procedure typically yields cultures that contain >90% neurons. All cell cultures were maintained at 37 °C in a humidified atmosphere containing 5% CO2 for 7 days prior to experiments. For in vitro validation, cultures were transduced with Cre-independent LV (LeGO-G-shRNA) harboring the same control or shGR sequences. Ninety-six hours post-infection, cells were stimulated with 100 nM dexamethasone for 4 h. For transduction efficiency tests, cells were visualized with a fluorescent/bright-field microscope. To evaluate knock-down efficiency primary cell cultures were harvested in Trizol (Invitrogen, Carlsbad, CA, USA). Total RNA (500–1000 ng) was reverse-transcribed into cDNA using Omniscript RT Kit (Qiagen). cDNA corresponding to 5–10 ng of total RNA was used for qPCR utilizing TaqMan Universal PCR Master Mix (Applied Biosystems, Foster City, CA, USA) and commercial TaqMan probes for Camk1g (Applied Biosystems; 4331182 Mm00460641_m1). Assays were run on the iCycler (Bio-Rad, Hercules, CA, USA). Data were calculated according to ΔCt method, using Actb or Hprt as reference genes. Fold change of DEX-induced gene expression was calculated and presented as a ratio over SAL-induced expression.

### 4.6. Stereotactic Surgery

C57BL/6J mice were anesthetized intraperitoneally with ketamine (7.5 mg/kg) and xylazine (1 mg/kg) before being placed in a stereotactic frame. Subcutaneous lidocaine (0.1 mL at 0.5%) at the level of the skull was injected prior to surgery to induce analgesia. Eye dehydration was prevented by topical application of the ophthalmic gel. The skin above the skull was shaved with an electric razor and disinfected with iodine solution before an incision was made. Viral vectors were injected with a microsyringe (10μL Hamilton syringe with a 30-gauge beveled needle) attached to a pump (UMP3-1, World Precision Instruments, Sarasota, FL, USA). To target the central amygdala, viral vectors were injected bilaterally using the following coordinates (from Bregma): posterior: −1.1 mm, lateral: ±2.5 mm, ventral: 4.6 mm, with an injection volume of 400 nL. The syringe was left in place for an additional 10 min after each injection to ensure complete diffusion of the vector. Mice were injected with LV-pSico-shCamk1g or LV-pSico-shGFP (control construct), and tested 3–4 weeks post-surgery. The transduction site of viral vectors in the CeA of mice used in the experiments was verified by immunohistochemistry.

### 4.7. Immunostaining of Transduced Brain Slices

Animals were perfused with 4% paraformaldehyde (PFA) buffered with phosphate-buffered saline (PBS), brains were postfixed in 4% PFA, and stored in PBS. Serial 40-μm-thick coronal sections from control and Camk1g KD mice obtained with vibratome were processed in parallel. Free-floating sections were rinsed in PBS, incubated with 10% normal goat serum in PBS with 0.2% Triton X−100 (PBS-Tx) for 90 min at room temperature (RT), and incubated overnight at RT with antibodies: anti-CaMKIγ (Invitrogen, Carlsbad, CA, USA; Anti Camk1g rabbit, Cat. No. PA5-28580) diluted in PBS-Tx containing 1% normal serum. The next day, sections were washed in PBS and incubated for 90 min at RT with fluorophore-conjugated secondary antibodies (A11073, Thermo Fisher), diluted 1:250 in PBS-Tx containing 1% normal serum. Afterward, sections were washed, incubated for 10 min at RT with Hoechst (1:2000 dilution in PBS, H3570, Thermo Fisher), washed, and embedded in ProLong Diamond Antifade Mountant (P36965, Thermo Fisher). Imaging was performed with a fluorescent confocal microscope (Nikon, Tokyo, Japan; model A1R). The contrast and brightness were altered for illustration purposes using the Fiji software. Co-expression of shCamk1g (GFP) and CaMKIγ was determined using Synchronize Windows and Cell Counter plugins.

### 4.8. Fear Conditioning

The cue fear conditioning procedure was performed as previously described [46]. Briefly, the procedure consisted of two parts: training and retrieval. Training consisted of the application of 5 tones (60 dB) paired with footshocks (0.4 mA, 1 min interval). The tone was presented for 20 s and each electric shock was administered during the last 2 s. During intershock intervals, freezing (immobility, except for respiratory movements) was measured as an expression of fear learning. At 24 h and 72 h after conditioning, mice were placed in a different context and were presented with the same tone for 3 min. Freezing was measured as an expression of cue-induced fear memory. The freezing data were recorded, stored, and analyzed using ANY-maze software (Stoelting, Wood Dale, IL, USA).

### 4.9. Light/Dark Box

During the light/dark box test, each mouse was placed in the middle of the dark compartment of the light/dark box. The apparatus consisted of two compartments (20 cm × 20 cm × 14 cm each), one of which was made of black Plexiglas lit by a dim light (50 lux). The other compartment was made of white Plexiglas and was illuminated with a lamp (300 lux). The 5 min trials were video-recorded and analyzed using ANY-maze software (Stoelting).

### 4.10. Y Maze

The Y-maze test was performed as previously described [47]. The apparatus consisted of three identical arms. For each mouse, the three arms of the Y-maze were randomly pre-assigned as the ‘start’ arm. The mouse was placed in that arm and allowed to freely explore the apparatus for 5 min. Mice exploratory behavior was assessed visually by scoring the successive entries into each of the three arms in overlapping triplet sets. Spontaneous alternations were calculated as the percentage ratio of the actual number of three successive different arms entries to the possible number of triplet sets (total arm entries minus two).

### 4.11. Statistical Analysis

Camk1g gene expression changes after dexamethasone injection and after stress were analyzed with Student’s *t* test using R 4.0.4 (R Foundation for Statistical Computing, Vienna, Austria). Fear conditioning measures were analyzed using two-way ANOVA with repeated measures for genotype and subsequent foot-shock factors in fear acquisition and for genotype and subsequent retrieval factors in retrieval using GraphPad Prism version 8.0.0 (GraphPad Software, San Diego, CA, USA). Light/dark box and Y maze measures were analyzed with Student’s *t* test using GraphPad Prism version 8.0.0.

## Figures and Tables

**Figure 1 ijms-23-12328-f001:**
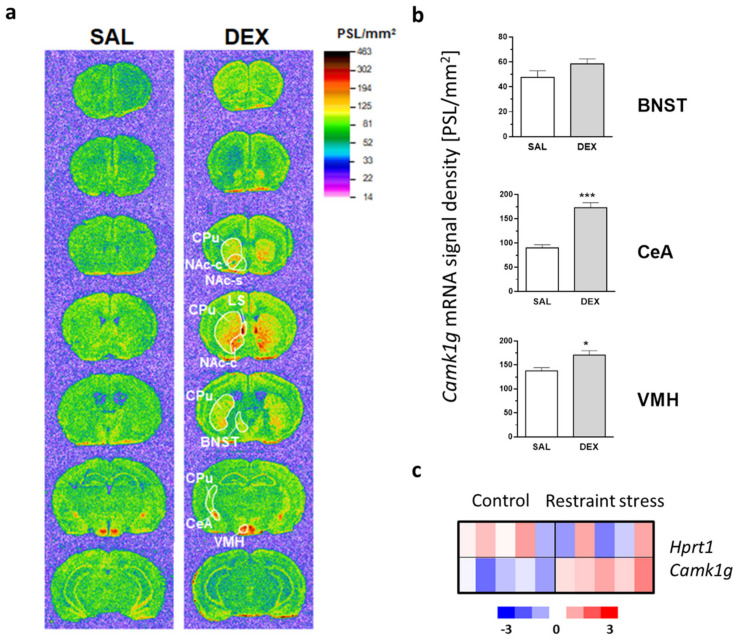
Camk1g distribution and regulation following dexamethasone (DEX) injection and restraint stress. (**a**) In situ hybridization autoradiograms depicting the brain distribution of Camk1g mRNA and its up-regulation by dexamethasone (DEX). The animals were sacrificed 4 h after the administration of DEX (16 mg/kg, i.p.) or saline (SAL). Optical densities of the autoradiograms, expressed in photostimulated luminescence units/mm^2^ (PSL/mm^2^), have been color-coded according to the scale in the upper right corner. The images are representative of 4 animals per group. (**b**) The plots show the results of a quantitative hybridization signal analysis in selected brain regions. The data, presented as mean ± SEM (*n* = 4), were analyzed by the unpaired two-tailed *t*-tests. *—*p* < 0.05; ***—*p* < 0.001. (**c**) Changes in Camk1g mRNA abundance levels in the amygdala of 5 mice per group after restraint stress compared with a housekeeping gene (Hprt1). The heatmap shows a z-scored abundance of log2-transformed counts for each gene as measured with microarray profiling. T-test *p*-value: Camk1g = 0.002; Hprt1 = 0.46. Abbreviations: BNST—bed nucleus of stria terminalis; CeA—central nucleus of amygdala; CPu—caudate/putamen; LS—lateral septum; NAc-c—nucleus accumbens core; NAc-s—nucleus accumbens shell; VMH—ventromedial hypothalamic nucleus.

**Figure 2 ijms-23-12328-f002:**
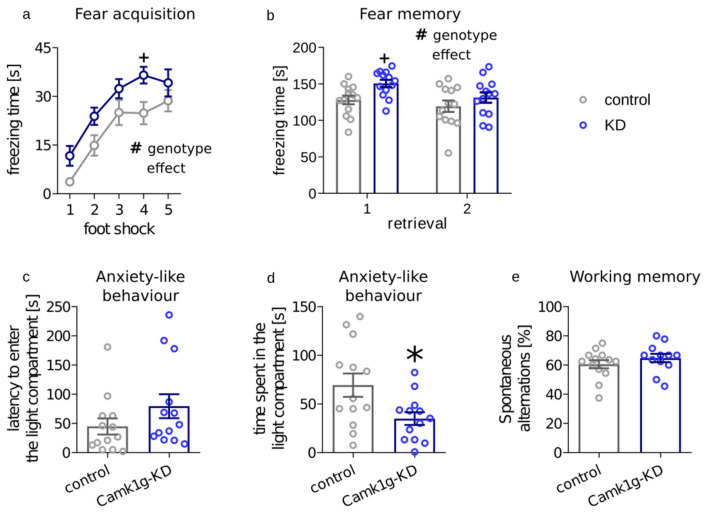
The behavioral effects of Camk1g knockdown in CeA. (**a**) Plot summarizing the freezing time of control (gray dots) and CeA-Camk1g-KD mice (blue dots) upon exposure to consecutive electric foot shocks. (**b**) Freezing behavior upon exposure to the cue during retrieval sessions performed at indicated sessions after the training. (**c**) Latency to enter the light compartment of the light-dark box (**d**) Total time spent in the illuminated compartment of the LD box (**e**) Spontaneous alterations measured in Y maze test. Mean ± SEM, * *p*  <  0.05 for genotype effect in *t* test, + *p*  <  0.05 for genotype effect in Bonferroni post hoc test, # *p*  <  0.05 for genotype effect in two-way ANOVA.

## Data Availability

Not applicable.

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
