# Peer review of "Glucocorticoid-Regulated Kinase CAMKIγ in the Central Amygdala Controls Anxiety-like Behavior in Mice"

_ijms, 2022, doi:10.3390/ijms232012328_

Round 1

Reviewer 1 Report

The authors present an interesting study on the role of CamKLg  in response to stress. They first demonstrate that the glucocorticoid Dexamethasone increases expression of Camk1g in specific brain areas. They then focus their attention to the amygdola and show that induced down-regulation of Camk1g alters the behavioural response to stress.

The authors  interpretation of the behavioural changes is that Camk1g  appears to be involved in reducing the ‘overshooting’ of the stress response. This is justified by the increased freezing behaviour of the knock down (KD)  mice following foot shock . However, the behaviour in the light/dark box could simply be interpreted as the KD mice having a higher baseline stress level. It would have been interesting to measure whether the KD treatment affected glucocorticoid levels in absence of stress. The authors may wish to comment on this point

Overall the data does indicate that the expression of Camk1g is an integral part of the response to stress and as such the finding represent an important contribution to the field

There are a few corrections needed:

Line 93 the scale is located  in the upper right corner (not lower left)

Line 96-98 only three plots are visible in the figure but seven are described in the figure legend

Line 129 (and other) The F values are difficult to interpret. They are not explained in the methods section. Indeed there should be a ‘Statistical Analysis’ section in the methods

Line 136-138 This sound confusing when first reading the statement while the results in the figure is clear: KD mice spent less time in the light  area which is aversive to anxious mice.

Line 140-141 It would be useful to quickly state the principle of the Y maze where the frequency of entering the least recently visited arm is an indication of a functioning working memory.

Line 147 the axis of panel b does not show times just 1 and 2

Line 149 presumed typo: should be alternation (not alteration)

Line 199 Something missing in this sentence

Line 203 Typo : for ( not fotr)

Line 324 Reference 14 does not seem to be the right reference for cue fear conditioning

Author Response

The authors would like to thank the reviewers for their insightful comments. Below is our point-by-point response.

rev1

The authors present an interesting study on the role of CamKLg  in response to stress. They first demonstrate that the glucocorticoid Dexamethasone increases expression of Camk1g in specific brain areas. They then focus their attention to the amygdola and show that induced down-regulation of Camk1g alters the behavioural response to stress.

The authors  interpretation of the behavioural changes is that Camk1g  appears to be involved in reducing the ‘overshooting’ of the stress response. This is justified by the increased freezing behaviour of the knock down (KD)  mice following foot shock . 

>>> However, the behaviour in the light/dark box could simply be interpreted as the KD mice having a higher baseline stress level. It would have been interesting to measure whether the KD treatment affected glucocorticoid levels in absence of stress. The authors may wish to comment on this point

RE: We have added new phrase to the discussion: How does this overshoot affect further glucocorticoid levels in the absence or during stress requires further investigation.

>>> Line 93 the scale is located  in the upper right corner (not lower left)

RE: The description was appropriately corrected.

>>> Line 96-98 only three plots are visible in the figure but seven are described in the figure legend

RE: We have now corrected the figure caption. As the same abbreviations are required for panel a and b, they were moved from subsection b to the end of the Figure description.

>>> Line 129 (and other) The F values are difficult to interpret. They are not explained in the methods section. Indeed there should be a ‘Statistical Analysis’ section in the methods

RE: The Statistical Analysis section was added to the Methods.

>>> Line 136-138 This sound confusing when first reading the statement while the results in the figure is clear: KD mice spent less time in the light  area which is aversive to anxious mice.

RE: The description was appropriately corrected.

>>> Line 140-141 It would be useful to quickly state the principle of the Y maze where the frequency of entering the least recently visited arm is an indication of a functioning working memory.

RE: The manuscript was appropriately corrected.

>>> Line 147 the axis of panel b does not show times just 1 and 2

RE: The figure caption was appropriately corrected.

>>> Line 149 presumed typo: should be alternation (not alteration)

RE: We have corrected the mistake.

>>> Line 199 Something missing in this sentence

RE: This phrase was changed to: “Despite  major limitations, the fear conditioning procedure used in the study is still regarded as a model for the stress-related pathogenesis of anxiety disorders [39]. Animals with Camk1g knockdown displayed higher freezing during acquisition and in the first response to a cue.”

>>> Line 203 Typo : for ( not fotr)

RE: This mistake has now been corrected.

>>> Line 324 Reference 14 does not seem to be the right reference for cue fear conditioning

RE: This reference was corrected to:

Tertil M, Skupio U, Barut J, Dubovyk V, Wawrzczak-Bargiela A, Soltys Z, Golda S, Kudla L, Wiktorowska L, Szklarczyk K, Korostynski M, Przewlocki R, Slezak M. Glucocorticoid receptor signaling in astrocytes is required for aversive memory formation. Transl Psychiatry. 2018 Nov 28;8(1):255. doi: 10.1038/s41398-018-0300-x. PMID: 30487639; PMCID: PMC6261947.

Reviewer 2 Report

The article: “Glucocorticoid-regulated kinase CAMKIγ in the central amygdala controls anxiety-like behavior in mice” is a nice demonstration of a molecular mechanism of anxiety. The work is well presented, and the experiments were performed properly. However, some issues must be addressed before publication.  

The main concerns are:

1.     Results section. The authors show the behavioral results in a very synthetic way, which makes their interpretation somewhat difficult. This section only shows the results of the ANOVAs, but not the post hoc analysis, which is essential to know which points (in the acquisition curve) and which groups (in the two memory recall tests) reach statistically significant differences. For example, in Figure 2b the authors say that there are statistically significant differences between the groups, but these are graphically identical.

Furthermore, in the figure caption (page 5 line 149) it is stated "Mean ± SEM, mean ± SEM, *p < 0.05, **p < 0.01, ***p < 0.001; for genotype effect in two-way ANOVA or in t test". However, there is no group marked with two or three asterisks or analyzed with the t test. These errors and omissions must be corrected.

2.     A Statistics section in the Methods detailing how data were analyzed would be very helpful. Please include it in the revised version.

3.     In the central nervous system, CAMKs are neuron-specific targets however, they are kinases that participate in several neural processes, such as including synapsis in terminal nerves, motility, axon growth, synthesis of aldosterone, and the cell cycle.  

In the absence of CAMKI, could the other CAMKs have redundant functions? If not, how could the authors explain the glucocorticoid regulation specificity? It can be explained in terms of its molecular structure?

Minor

Figure legend 1.

Line 91-93: Optical densities of the autoradiograms, ex-91 pressed in photostimulated luminescence units/mm2 (PSL/mm2), have been color-coded according to the scale in the lower left corner. The scale is in the lower right corner of the letter a.

Supplementary figure 3. Left panel. The image of the illustrative brain atlas must be improved. 

Author Response

The authors would like to thank the reviewers for their insightful comments. Below is our point-by-point response.

rev2

The article: “Glucocorticoid-regulated kinase CAMKIγ in the central amygdala controls anxiety-like behavior in mice” is a nice demonstration of a molecular mechanism of anxiety. The work is well presented, and the experiments were performed properly. However, some issues must be addressed before publication.  

The main concerns are:

>>> 1.     Results section. The authors show the behavioral results in a very synthetic way, which makes their interpretation somewhat difficult. This section only shows the results of the ANOVAs, but not the post hoc analysis, which is essential to know which points (in the acquisition curve) and which groups (in the two memory recall tests) reach statistically significant differences. For example, in Figure 2b the authors say that there are statistically significant differences between the groups, but these are graphically identical.

Furthermore, in the figure caption (page 5 line 149) it is stated "Mean ± SEM, mean ± SEM, *p < 0.05, **p < 0.01, ***p < 0.001; for genotype effect in two-way ANOVA or in t test". However, there is no group marked with two or three asterisks or analyzed with the t test. These errors and omissions must be corrected.

RE: Both the Figure caption and text were updated and now include Bonferroni post hoc test results.

>>> 2.     A Statistics section in the Methods detailing how data were analyzed would be very helpful. Please include it in the revised version.

RE: We have added a Statistical Analysis section in the Methods.

>>> 3.     In the central nervous system, CAMKs are neuron-specific targets however, they are kinases that participate in several neural processes, such as including synapsis in terminal nerves, motility, axon growth, synthesis of aldosterone, and the cell cycle.  In the absence of CAMKI, could the other CAMKs have redundant functions? If not, how could the authors explain the glucocorticoid regulation specificity? It can be explained in terms of its molecular structure?

RE: New paragraph was added to the Discussion:

“In the central nervous system, CAMKs play a crucial role in several neural processes, such as mobilization of synaptic vesicles, modulation of ion channels, regulation of gene expression, regulation of muscle contraction, and LTP (Zalcman, 2018). Despite not much being known about substrate specificity of particular CAMKs, there are studies showing that such phenomena exist. For example, activity-dependent cap-dependent translation, but not a cap-independent translation, is sensitive to knockdown of CaMKIγ and CaMKIβ but not to knockdown of CaMKIα, CaMKIδ, or CaMKII  (Srivastava, 2012). Glucocorticoids, through this mechanism, could enable activity-dependent translation in particular neuronal circuits.”

Minor

Figure legend 1.

>>> Line 91-93: Optical densities of the autoradiograms, ex-91 pressed in photostimulated luminescence units/mm2 (PSL/mm2), have been color-coded according to the scale in the lower left corner. The scale is in the lower right corner of the letter a.

RE: The figure caption is now corrected.

>>> Supplementary figure 3. Left panel. The image of the illustrative brain atlas must be improved.

RE: The Supplementary figure has been corrected.
